# BaTiO_3_ Functional Perovskite as Photocathode in Microbial Fuel Cells for Energy Production and Wastewater Treatment

**DOI:** 10.3390/molecules28041894

**Published:** 2023-02-16

**Authors:** Noureddine Touach, Abdellah Benzaouak, Jamil Toyir, Youssra El Hamdouni, Mohammed El Mahi, El Mostapha Lotfi, Najoua Labjar, Mohamed Kacimi, Leonarda Francesca Liotta

**Affiliations:** 1Laboratory of Spectroscopy, Molecular Modelling, Materials, Nanomaterials, Water and Environment, Environmental Materials Team, ENSAM, Mohammed V University in Rabat, Avenue des Forces Armées Royales, Rabat B.P. 6207, Morocco; 2Laboratoire des Procédés, Matériaux et Environnement (LPME), Faculté Polydisciplinaire (FP-Taza), Faculté des Sciences Et Techniques de Fès (FST-Fès), Université Sidi Mohammed Ben Abdellah, Taza B.P. 1223, Morocco; 3Laboratory of Physical Chemistry of Materials, Catalysis and Environment, Department of Chemistry, Faculty of Sciences, Mohammed V University in Rabat, Rabat B.P. 1014, Morocco; 4Istituto per lo Studio dei Materiali Nanostrutturati (ISMN)-CNR, via Ugo La Malfa, 153, 90146 Palermo, Italy

**Keywords:** microbial fuel cell, wastewater treatment, BaTiO_3_, photocathode, perovskite, chemical oxygen demand

## Abstract

Microbial fuel cells (MFCs) provide new opportunities for the sustainable production of energy, converting organic matter into electricity through microorganisms. Moreover, MFCs play an important role in remediation of environmental pollutants from wastewater with power generation. This work focuses on the evaluation of ferroelectric perovskite materials as a new class of non-precious photocatalysts for MFC cathode construction. Nanoparticles of BaTiO_3_ (BT) were prepared and tested in a microbial fuel cell (MFC) as photocathode catalytic components. The catalyst phases were synthesized, identified and characterized by XRD, SEM, UV–Vis absorption spectroscopy, P-E hysteresis and dielectric measurements. The maximum absorption of BT nanoparticles was recorded at 285 nm and the energy gap (Eg) was estimated to be 3.77 eV. Photocatalytic performance of cathodes coated with BaTiO_3_ was measured in a dark environment and then in the presence of a UV–visible (UV–Vis) light source, using a mixture of dairy industry and domestic wastewater as a feedstock for the MFCs. The performance of the BT cathodic component is strongly dependent on the presence of UV–Vis irradiation. The BT-based cathode functioning under UV–visible light improves the maximum power densities and the open circuit voltage (OCV) of the MFC system. The values increased from 64 mW m^−2^ to 498 mW m^−2^ and from 280 mV to 387 mV, respectively, showing that the presence of light effectively improved the photocatalytic activity of this ceramic. Furthermore, the MFCs operating under optimal conditions were able to reduce the chemical oxygen demand load in wastewater by 90% (initial COD = 2500 mg L^−1^).

## 1. Introduction

Sustainable development is crucial for addressing environmental and societal challenges such as climate change, population growth, and resource reduction. Nowadays, it is more necessary than ever to adopt highly efficient methods that take into account economic, social and environmental aspects, such as reducing emissions from fossil energy, promoting renewable energy and protecting biodiversity. In the current conjuncture of climate change and water crises, the development of technologies for the conversion of renewable energies, particularly the chemical energy of effluents into bioelectricity [1,2,3], has become, more than ever, a necessity. In this context, the microbial fuel cell technology appears as a promising solution, capable of wastewater treatment and generating bioelectricity through the actions of microorganisms [4,5]. The large-scale implementation of this technology is directly related to the economic cost and efficiency of all the components of this system, in particular the cathode [6,7,8]. The design of an efficient air cathode for oxygen reduction (ORR) in microbial fuel cells (MFCs) is a crucial step in the building process of this technology [5,9]. The most active systems are mainly based on noble metals, in particular Pt, Rh and Pd. However, they are more and more expensive and difficult to access in the market, which can be a significant disadvantage when MFC devices come to be commercialized [10,11]. Thus, recently, the development is focused on the implementation of catalysts, based on non-precious metals, that are active, stable and less expensive [12]. However, despite this considerable progress, the activity and sustainability of tested catalysts for ORR are still insufficient.

Important electrode reactions including oxygen reduction reaction (ORR) and oxygen evolution reaction (OER) play a key role in driving the operating processes of air cathode biofuel cells. Inorganic perovskite oxides have been reported as effective systems for both ORR and OER [13,14]. Interestingly, oxygen-deficient perovskite catalysts such as BaTiO_2.76_ exhibit high catalytic activity simultaneously for the ORR and the OER. In these processes, surface and bulk oxygen diffusion through the solid is an important issue which can be improved using defective structure perovskites. These materials are well known to possess remarkable electronic and magnetic properties, in addition the presence of oxygen vacancies allowing high mobility of oxygen ions through the lattice [15,16]. 

Besides their ability to ensure an excellent mobility of oxygen ions, suitably prepared nanostructured perovskites can perform well as catalysts for ORR with low energy for oxygen activation and high electron transfer kinetics, resulting in a good electrocatalytic activity [17]. Recently, the potential of some ferroelectric perovskite electrodes has been investigated through electrocatalytic and photoelectrocatalytic tests carried out in single-chamber MFCs. Such perovskites were able to enhance the performance of MFCs and the energy recovery. It was reported that LiNbO_3_ cathodes exhibited a maximum power of 131 mW m^−3^ under irradiation and a maximum chemical oxygen demand (COD) removal of 84% [9]. In another study, LiTaO_3_, a ferroelectric material, has shown good performances as a photocathode with a maximum generated power of 55 mW m^−3^ and a COD removal power of 66% [18]. The electrocatalytic performance of ceramic materials, such as (Li_0_._95_Cu_0.15_)Ta_0.76_Nb_0.19_O_3_ and Li_0.95_Ta_0.76_Nb_0_._19_Mg_0.15_O_3_, has also been investigated when used as photocathodes for MFCs. They can generate maximum powers of 19.77 mW m^−3^ (25.13 mW m^−2^) and 228 mW m^−2^, respectively [19,20,21] and percentages of 93% and 95.70% in terms of COD removal, thus showing good potential as photocathodes in MFC devices.

Among other ferroelectric perovskites, it is worth mentioning BaTiO_3_, a functional ceramic, recognized for its ferroelectric properties resulting from distorted unit cells and the displacement of the positive and negative charge barycenter of the crystal [22]. Recently, BaTiO_3_ as a catalyst has attracted much attention, particularly for the reduction of CO_2_ to methanol [23], partial oxidation of methane [24], ammonia synthesis and toluene decomposition [25,26]. BaTiO_3_ has also been widely studied in photocatalysis thanks to its internal electric field generated by permanent polarization [27,28] that enhances the separation of photo-induced charge carriers [29,30]. This material has been tested in several photocatalytic reactions, especially the degradation of organic dyes under white visible light [31,32,33,34]. According to the literature, the following reactions are the key steps of photo-induced reactions catalyzed by BaTiO_3_ [30,35]: 

The absorption of photons by barium (hν ≥ E_bg_ = 3.23 eV), Equation (1):(1)$${\mathrm{BaTiO}}_{3}\overset{h\nu }{\to }{\;\mathrm{BaTiO}}_{3}\left(q^{-}+q^{+}\right)$$

The negative charges (q^−^) can be rapidly trapped by molecular oxygen on the surface to form superoxide radical (O_2_^∙−^), as shown in chemical Equation (2):(2)$$O_{2}+{\;q}^{-}\;⟶{\;O}_{2}^{∙-}$$

This radical could be combined with H^+^ coming from PEM to form H_2_O Equation (3):(3)$$O_{2}\;\overset{e^{-}}{\to }{\;O}_{2}^{∙-}\overset{e^{-}}{\to }{\;H}_{2}O_{2}\overset{e^{-}}{\to }{\;\mathrm{OH}}^{∙}+{\;\mathrm{OH}}^{-}\overset{e^{-}}{\to }\;2{\mathrm{HO}}_{2}\;$$

On these bases, BaTiO_3_ (BT) appears a very suitable material for electrocatalytic and photocatalytic applications. In the present study, we investigate, for the first time, the catalytic performance of BaTiO_3_ as a cathode for MFC devices, both in the presence and absence of UV–Vis light irradiation. The photocatalytic activity of this ferroelectric material, in terms of energy performance and wastewater purification, was studied in a single-chamber MFC fed with dairy industry and domestic wastewater. 

## 2. Results and Discussion

### 2.1. Characterization of BaTiO_3_ Structure and Morphology

The X-ray diffraction (XRD) technique was used to verify the purity of the synthesized phase and determine the crystalline structure of the BaTiO_3_ material. The diffraction pattern is shown in Figure 1. All the peaks are in good agreement with the reference, tetragonal perovskite structure with P4mm space group 3 (JCPDS No. 76–0744). Tetragonality can be confirmed by the peaks at specific angles, such as 22.3°, 31.6°, 38.9°, 45.3°, 50.9°, 54.1°, 56.2°, 65.9° and 70.3° 2θ. Such peaks were attributed to the (100), (101), (111), (200), (201), (211), (202), (212), (310) and (113) crystal planes, respectively. These observations are consistent with previous studies [30,36].

Scanning electron microscopy (SEM) analyses were performed for detailed insights into the morphology of the prepared BaTiO_3_ sample. Figure 2a,b illustrates SEM images of the BT powder. As results from this microstructure analysis, the particles having nanometric sizes between 40 and 200 nm exhibited the crystalline shape of tetragonal parallelepipeds in agreement with XRD results. Furthermore, the elemental mapping SEM of Ba, Ti and O in Figure 2c–e indicates the homogeneous dispersion of these elements. The energy dispersive spectroscopy (EDS) results are illustrated in Figure 2b. From this analysis, the chemical composition of the prepared sample was revealed with high accuracy. The ratio between the atomic percentages of Ba (18.5%), Ti (16.4%) and O (65.1%) is equal to 1.12:1:3.5, which is near the stoichiometric ratio of BaTiO_3_.

It is worth noting that BaTiO_3_, when is subjected to high temperatures or strong electric fields, can undergo distortion of the crystalline structure, changing its electrical and electronic properties. This distortion can lead to a change in polarization, which is related to the appearance of free electrical charges on the surface of the ceramic. This polarization change is responsible for the ferroelectricity of BaTiO_3_ that is one of the most important properties of the material, particularly in catalysis [37].

The structural distortion of BaTiO_3_ can also cause a change in the electronic energy bands, which can affect the electronic transport properties of the material. It might lead to an increase in the density of electronic states on the surface of the ceramic, improving the electrical conductivity. 

In this regard, dielectric measurements for the prepared material (BT) were performed. The P-E hysteresis loop and dielectric response of the BT ceramic are plotted in Figure 3. As can be seen in Figure 3a, the material presents a typical P-E hysteresis loop at room temperature, with a remanent polarization value (Pr) of approximately 1.807 μC cm^−2^. It has been pointed out that the loop area of the material is relatively large, which confirms the relatively high dielectric constant. In addition, the dielectric response of the BT is also an important property, as it is related to its ability to store electrical energy. Based on Figure 3b, plotting the dielectric constant as a function of the temperature, BaTiO_3_ exhibits phase transitions from an orthorhombic phase to a tetragonal phase at 22 °C and from a ferroelectric tetragonal phase to a paraelectric cubic phase at 132 °C, which suggests that the BaTiO_3_ is a typically ferroelectric material with remanent polarization along the [001] axis at room temperature [37,38].

### 2.2. Optical Properties

The absorption spectrum of the BaTiO_3_ material was measured to provide information about its electronic structure. The absorption edge recorded at 285 nm is due to the electronic transitions between the valence and the conduction bands of the studied material. This transition is known as an interband transition and indicates the band gap of the BT. It should be noted that the band gap of the material is an essential property because it is related to the energy required for an electron to be excited from the valence band to the conduction band. A larger band gap indicates higher energy required for excitation, which can affect the photocatalytic activity of the material.

Tauc’s plot, also shown in Figure 4, is a graphical representation of the absorption coefficient of the material as a function of the photon energy. Generally, the intersection with the x-axis of the linear part of this curve gives access to the band gap of the material. The Tauc’s plot of the BaTiO_3_ material confirms that the band gap is around 3.77 eV, which is consistent with the value obtained from the absorption spectrum and comparable to what was found in a previous work [39]. 

Thus, the optical properties of the BT material indicate that it has a band gap of 3.77 eV, which suggests that the material would be active in the UV light region. This makes the material suitable for various applications such as photocatalysis, solar cells and optoelectronics. However, it should be noted that the material’s optical properties alone are insufficient to fully assess its photocatalytic activity. Further studies, such as on photocatalytic activity, were performed in an air-cathode MFC to confirm this.

### 2.3. Electrochemical Characterization

Cyclic voltammetry (CV) was employed to investigate the nature of the processes occurring at the surface of the electrode and at the interface with the solution (denoised water or wastewater). The characteristics of the mixture of dairy industry and domestic wastewater used as a feedstock for the MFCs are listed in Materials and Methods, Section 3.

Figure 5a,b exhibit the voltammograms of the BT material with a scan rate variation from 20 mV s^−1^ to 100 mV s^−1^. We observed an increase in the redox current intensity by increasing the scan rate, in accordance with the literature [40,41]. Furthermore, the BT-based electrode demonstrated a larger electroactive zone in wastewater than in the denoised medium, suggesting a potential application for wastewater treatment and power generation in MFC devices. The voltammogram size of BT is in agreement with other works when graphite electrodes were used in MFC systems for wastewater treatment [6,42].

### 2.4. Performance in Single-Chamber MFC

Experiments were performed to analyze the photoelectrocatalytic activity of BaTiO_3_, using it for the first time in an MFC as a photocathode. Figure 6a,b show the power and polarization curves of an MFC equipped with a cathode coated with BT, after 120 h in operation and in the presence and absence of light conditions. When the BT-based cathode was exposed to UV–visible light, the open circuit voltage (OCV) of the MFC system increased from 280 mV to 387 mV. The dependence of the polarization curve of the MFC is illustrated in Figure 6a. Both graph plots (BT with/without light source) show three potential loss zones, generally observed for MFCs, which are the activation loss, the ohmic loss and the concentration loss [43].

Power density data were a key parameter for evaluating MFC system behavior. The MFC with a BT-irradiated cathode presents the best MFC performance with a maximum power density of 498 mW m^−2^ associated with a current density of 2408 mA m^−2^, which is 7.8 times higher than that of an MFC functioning with BT without irradiation (64.02 mW m^−2^, 1440 mA m^−2^) (Figure 6b). This power performance is due to the photoelectrocatalytic activity of the BaTiO_3_ material. These results are very encouraging compared to the various Pt-based cathodes recognized as potential catalysts used in microbial fuel cells [12,44,45].

Recently, various ferroelectric photocatalytic materials have also been applied as cathodes in MFCs. BiFeO_3_-based cathodes, in the form of nanoparticles synthesized by the hydrothermal method, achieved a maximum power output of 332 mW m^−2^ [46], that is lower than the value measured for BaTiO_3_-based cathodes studied in this work. On the other hand, cathodes based on non-stoichiometric ferroelectric materials such as iso-type LiMO_3_ (M = Nb, Ta) modified by Cu^2+^, Mg^2+^ were investigated [19,20,21]; their performance as photocatalysts in air-cathode MFCs was inferior to that of the present BaTiO_3_ material. Moreover, it is worth mentioning that other types of TiO_2_ cathode, the most well-known photocatalyst, led to a power output in this device in the (4.34–239) mW m^−2^ range, which also remains significantly lower than that obtained with the BaTiO_3_-based configuration [47,48,49,50,51]. 

In the MFC, at the bioanode, the degradation of organic substrate can generate bioelectrons (e^−^) and inorganic matter through active microorganisms, taking as an example the oxidation of acetate (as shown in Equation (4)). These electrons accumulated in the bioanode are then transferred by an external charge driven by the potential difference in the MFC to reach the photocathode, and protons migrate to the photocathode surface via a proton exchange membrane (as shown in Equation (6)) where photoelectrocatalysis process occur.

In this study, the photocathode was coated with BaTiO_3_ nanoparticles. When exposed to light, the electrons in the valence band of BaTiO_3_ are excited to the conduction band, while the holes are retained in the valence band (as shown in Equation (5)). Additionally, holes can combine with electrons from the anode to generate an electric current (as shown in Equation (7)) and decrease the rate of recombination of electron–hole pairs.
(4)$${\mathrm{CH}}_{3}{\mathrm{COO}}^{-}+2H_{2}O\;\to 2{\mathrm{CO}}_{2}+7H^{+}+8e^{-}$$
(5)$${\mathrm{BaTiO}}_{3}+h\nu \;\to e_{\mathrm{cb}}^{-}+{\;h}_{\mathrm{vb}}^{+}$$
(6)$$4H^{+}+{\;O}_{2}+4e/e_{\mathrm{cb}}^{-}\;\to \;2H_{2}O$$
(7)$$h_{\mathrm{vb}}^{+}+{\;e}^{-}\;\to \mathrm{recombination}$$

In addition, the spontaneous polarization in this material, along the c-axis, is due to a small ionic displacement mostly dominated by Ti ion displacement with respect to the oxygen [27,28]. This can improve the transport of photoinduced electron–hole charge carriers to the catalyst interfaces that were separated by the spontaneous polarization. Ferroelectric/electrode interfaces in any metal/semiconductor contact are known to generate defects which may change redox activity at the junction between deposited material and the support of the electrode [52]. A large number of charged defects are usually observed at the near surface of a ferroelectric-type material such as BaTiO_3_ as, for many kinds of semiconductive metal, oxides may change the structure and polarization of ultrathin BaTiO_3_-coated cathodes. Therefore, the ORR can be promoted at the positively charged surface of the material.

COD removal efficiency is an important indicator of MFC performance in terms of treatment efficiency and energy production. Catalysts with better ORR performance require more electrons for oxygen reduction reactions. Accordingly, MFCs are expected to actively decompose organic substrates. The results of the present study indicated that all MFCs equipped with a BT-coated cathode had high COD removal efficiency in the presence and absence of light (Table 1). When the BT-based cathode was exposed to UV–visible light, a significant COD reduction was obtained, reaching 90%. However, the non-irradiation MFC system only achieved 74.5% COD removal. This COD removal value demonstrates a good correlation with the maximum power density results that increased from 64 mW m^−2^ to 498 mW m^−2^ in the absence and presence of UV–visible light, respectively. 

Thus, we have demonstrated that the ferroelectric BaTiO_3_ photocathode is a promising material for use in MFCs due to its ability to reduce COD and generate electricity.

## 3. Materials and Methods

### 3.1. Catalyst Preparation and Characterization

BaTiO_3_ (BT) powder was prepared following the solid-state synthesis by firing at high temperature a mixture of BaCO_3_ (Merck, Darmstadt, Germany, 99%) and TiO_2_ (Merck, Germany, 99%) (as shown in Equation (8)). The processing steps were: ball milling, calcining at 600 °C for 12 h to expel CO_2_, then increasing to 800, 1000 and 1100 °C for 12 h. The heat treatment was followed by slow cooling. The synthesized products were typically fine powders.
(8)$${\mathrm{BaCO}}_{3}+{\;\mathrm{TiO}}_{2}\;⟶{\;\mathrm{BaTiO}}_{3}+{\;\mathrm{CO}}_{2}\left(600\;{}^{\circ}C\leq T\leq 1100\;{}^{\circ}C\right)$$

The crystalline structure of the BaTiO_3_ was investigated by XRD analysis in the angular range 10–80° 2θ, by using a 0.02° step size and a scan speed of 0.05°/s. Diffraction patterns were registered with a Bruker D8 Advance diffractometer (Germany) equipped with monochromatic CuKα radiation (1.54 Å). 

The morphology, composition and elemental mapping distribution of the BaTiO_3_ compound were analyzed by scanning electron microscopy (SEM) and energy dispersive X-ray (EDX) analysis on QUATTRO S-FEG-Thermo Fisher equipment. 

The powder of BaTiO_3_ (BT) material was pressed (8 tons) into the form of discs, and then it was sintered at 1100 °C. The obtained pellet was metallized by silver lacquer and then the measurements of the dielectric characteristics were carried out using an impedance analyzer (Agilent 4294A) in the frequency range of 1Hz–35 KHz with an applied voltage of 500 mV in a temperature range of 20–200 °C.

The study of the optical properties was conducted through UV–visible–NIR diffuse reflectance spectroscopy at room temperature, covering the wavelength range of 190–1000 nm. A Varian Cary 5-E spectrometer (Australia) was used, equipped with an integrating sphere coated with polytetrafluoroethylene (PTFE) and a double monochromator.

The energy gap (E_g_) was estimated through performing the UV–Vis absorption analysis. The band gap of BaTiO_3_ photocathode was calculated based on the Wood–Tauc method following Equation (9) [53,54,55]:(9)$${\left(\alpha h\nu \right)}^{2}=A{\left(h\nu -E_{g}\right)}^{n}$$
where α represents the absorption coefficient, h and ν are the Planck constant and the incident light frequency, respectively, A is a constant. The factor n in the equation is dependent on the type of electron transition and is either equal to 1/2 or 2 for direct and indirect band gap transitions, respectively.

The electrochemical measurements were performed using a potentiostat (SP-150-BioLogic science instruments, French, Grenoble) equipped with EC-Lab software. The system contained three electrodes, specifically a working electrode, a reference electrode based on saturated calomel and a counter-electrode in the form of a platinum wire. The influence of scan rate variation was examined using cyclic voltammetry in denoised water and wastewater.

### 3.2. MFC Operation

The synthesized BaTiO_3_ ceramic were evaluated in air-cathode MFCs constituted by single reactors of 250 mL anode capacity and equipped with an external jacket, to control the operating temperature at 27 °C. The cathode was composed of a mixture of the synthesized BaTiO_3_ phases as catalysts and a solution of polytetrafluoroethylene (PTFE) (solid, Sigma-Aldrich, Burlington, MA, USA) as binder. The mixture was mechanically pressed onto a piece of carbon cloth with a surface area of 1 cm^2^. The total mass was 60 mg/cm^2^ in a 1:9 mass ratio of PTFE:catalyst. The anode consisted of 100 g of graphite granules (Graphite Store, IL, USA) with a diameter of 2–6 mm and a graphite rod (Graphite Store, USA) with a diameter of 3.18 mm, which was connected to the cathode with an external resistance of 1 kΩ. The anode chamber was filled with a mixture of dairy industry and domestic wastewater (125 mL) whose characteristics are listed in Table 2. The reactors were installed with a Nafion-type proton exchange membrane with a diameter of 3.8 cm located between the air-cathode and the anode chamber [56]. The experiment was performed for 120 h, under UV–visible light irradiation using a 500W power lamp.

The polarization and power measurements were obtained from the variation of different external resistors (11 MΩ–1 Ω). The open circuit voltages (OCVs) of the MFCs were measured with a voltmeter. The current (I) and power (P) densities were obtained by the equations: I = V/R and P = V^2^/R, where V is the cell voltage and R is an external resistance, normalized to the surface of the cathode.

### 3.3. COD Removal Efficiency

The chemical oxygen demand (COD) was determined using the APHA standard method [57] on a photoLab 7600 UV–visible spectrophotometer (WTW, Germany) and determined with Equation (9):(10)$${\%\;\mathrm{COD}}_{r}=\frac{{\left[\mathrm{COD}\right]}_{i}-{\left[\mathrm{COD}\right]}_{f\;}}{{\left[\mathrm{COD}\right]}_{i}}\times 100$$
where [COD]_i_ (mg/L) = initial COD charge in the wastewater inlet to the anode chamber and [COD]_f_ (mg/L) = COD of the effluent in the anode chamber at the end of the experiment.

## 4. Conclusions

This work investigated for the first time the possible application of BaTiO_3_ (BT) as a cathode material for microbial fuel cells (MFCs), in the presence and absence of a UV–Vis light source, using a mixture of dairy industry and domestic wastewater as feedstock.

The energy performance of BT was significantly improved when the MFC was working under light conditions. The maximum power densities generated by the MFC using BT-based cathodes increased from 64 mW m^−2^ to 698 mW m^−2^ in the absence and presence of UV–visible light, respectively. The BT-based MFC also exhibited a remarkable COD removal performance, with a maximum of 90% obtained in the presence of a light source.

The BT compares well with other alternative ferroelectric cathode materials reported in the literature for MFC devices.

In conclusion, this work demonstrated the feasibility of a sustainable process for energy production, by using as cathodic material, in MFCs, BaTiO_3_ that is a functional and cost-effective ferroelectric perovskite, able to convert organic matter into electrical energy while simultaneously purifying water. 

## Figures and Tables

**Figure 1 molecules-28-01894-f001:**
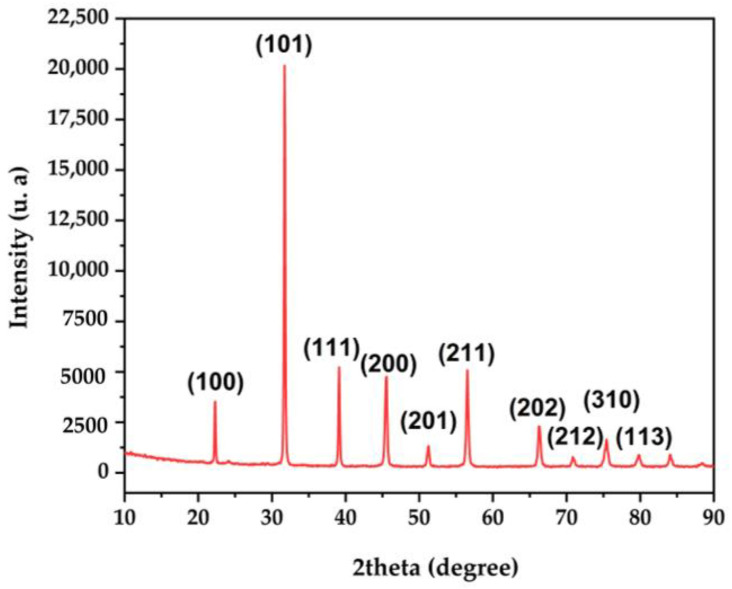
X-ray diffraction pattern of BaTiO_3_ powder.

**Figure 2 molecules-28-01894-f002:**
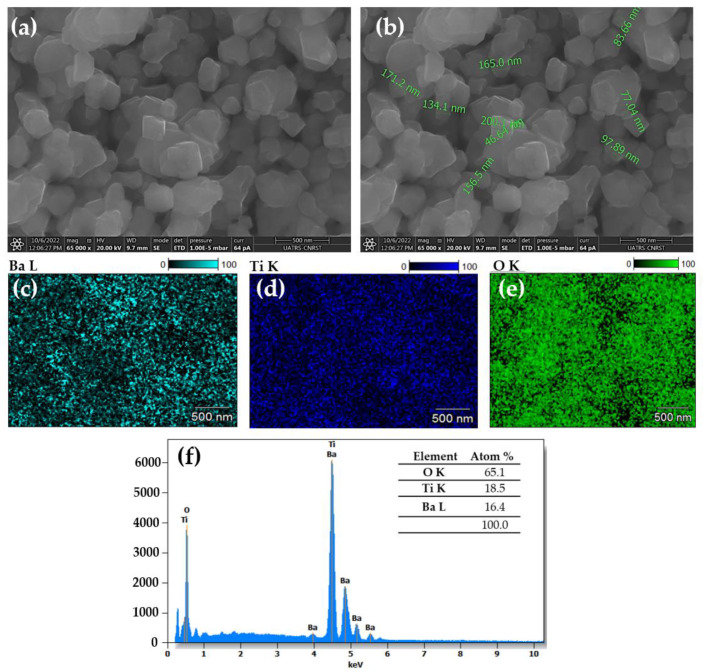
SEM images of BaTiO_3_ powder (**a**,**b**), elemental mappings (**c**–**e**), EDS spectrum (**f**).

**Figure 3 molecules-28-01894-f003:**
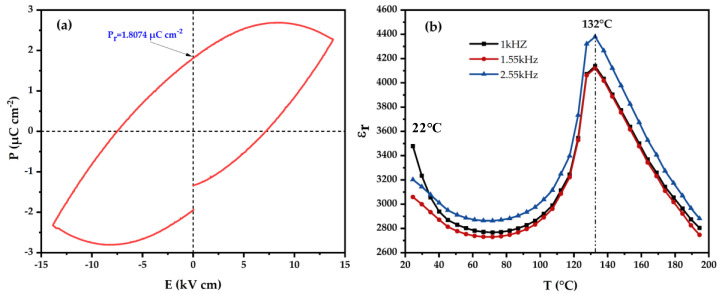
(**a**) P-E hysteresis loop of BaTiO_3_. (**b**) Temperature dependences of dielectric response of BaTiO_3_ at 1 KHz, 1.55 KHz and 2.55 KHz.

**Figure 4 molecules-28-01894-f004:**
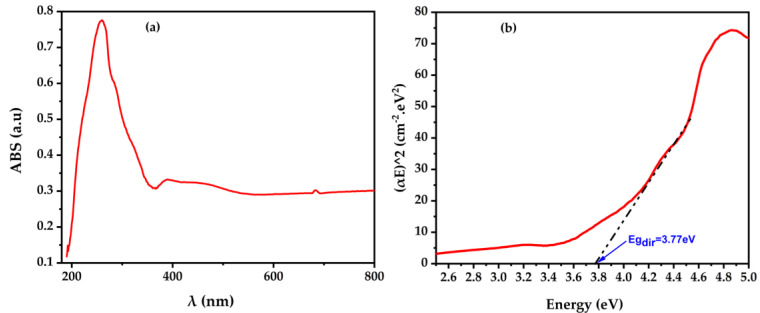
UV–visible absorbance of BaTiO_3_ (**a**) and Tauc’s plot (**b**).

**Figure 5 molecules-28-01894-f005:**
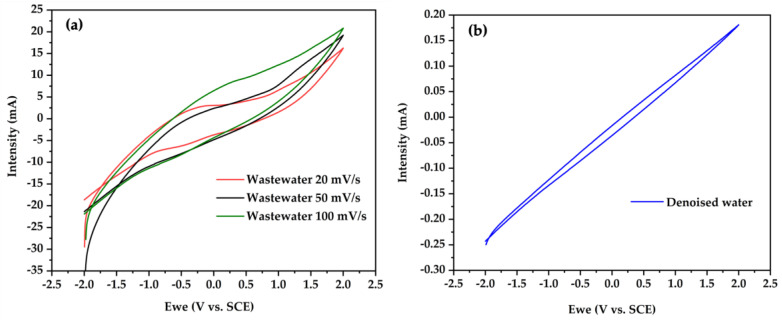
Cyclic voltammograms of BaTiO_3_ catalyst at different scan rates in (**a**) wastewater and (**b**) denoised water.

**Figure 6 molecules-28-01894-f006:**
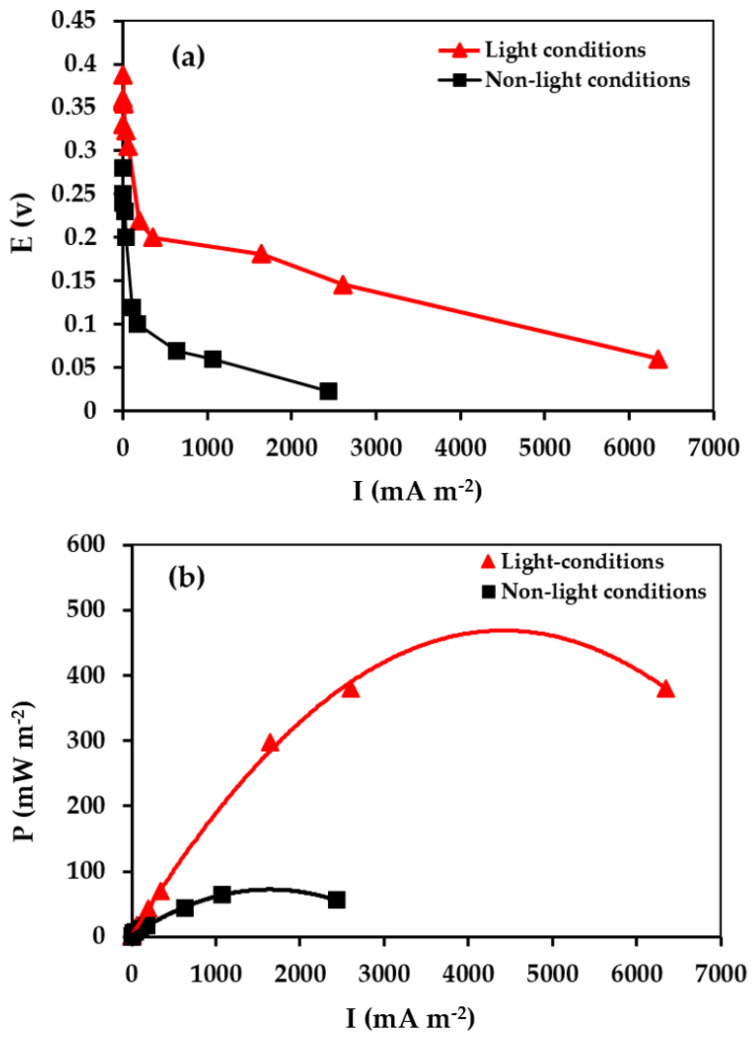
(**a**) Polarization curves (**b**) and power density for BaTiO_3_.

**Table 1 molecules-28-01894-t001:** Performance of MFCs equipped with BaTiO_3_ ceramic cathode in the presence and absence of light.

BaTiO_3_ Sample	^a^ Eg (eV)	P_max_ (mW m^−2^)	I at P_max_ (mA m^−2^)	OCV (mV)	^b^ COD_r_ after 120 h (%)
Absence of light	-	64.0	1440	280	74.5
Presence of light	3.77	498.0	2408.5	387	90

^a^ Eg (eV): Energy gap. ^b^ COD_r_: Chemical oxygen demand removal.

**Table 2 molecules-28-01894-t002:** The physico-chemical parameters of wastewater before testing in single-chamber MFC.

	COD (mg L^−1^)	^a^ BOD (mg L^−1^)	Dissolved Oxygen (mg L^−1^)	^b^ EC (µS cm^−1^)	pH	Temperature (°C)
Wastewater tested	2500	440	0.8	4330	7.8	27

^a^ BOD: Biochemical oxygen demand. ^b^ EC: Electrical conductivity.

## Data Availability

All the data are included in the article.

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
