# Peer review of "BaTiO3 Functional Perovskite as Photocathode in Microbial Fuel Cells for Energy Production and Wastewater Treatment"

_molecules, 2023, doi:10.3390/molecules28041894_

Round 1

Reviewer 1 Report

I have outlined below my understanding and thoughts on it, hoping they will help further improve its scientific contribution.

(1) In the abstract:

-          Add more quantitative data.

 (2): In the results:

-          Need to compare the results in Section 2.2 with more recent studies in the literature.  

-           

(3): In the Methodology:

-          Need to have references in section 3.1.  

Author Response

We are very grateful for the time you took to read the manuscript and for your valuable suggestions to improve it. Herein are responses point by point to the comments:

 (1) In the abstract:

-          Add more quantitative data.

Responses: The abstract was modified accordingly.

 (2): In the results:

-          Need to compare the results in Section 2.2 with more recent studies in the literature.

Responses: The Tauc's plot of the BaTiO3 material confirms that the band gap is around 3.77 eV, which is consistent with the value obtained from the absorption spectrum and comparable to other value found in previous works [ACS Omega 2019 4 (6), 9673-9679].

 (3): In the Methodology:

-          Need to have references in section 3.1.

Responses: The needed references were added in the text:

 52- Weak absorption tails in amorphous semiconductors. Physical review B 1972, 5, 3144;

 53- Optical properties and electronic structure of amorphous germanium. physica status solidi 1966, 15, 627-637;

 54- Evaluation of the Tauc method for optical absorption edge determination: ZnO thin films as a model system. Physica status solidi 2015, 252, 1700-1710.

Reviewer 2 Report

The manuscript titled "BaTiO3 functional perovskite as photocathode in microbial fuel cells for energy production and wastewater treatment" describes the use of BaTiO3 as photocathode catalytic components for the removal of COD from wastewater. The manuscript discusses about the synthesis and characterization of the functional perovskite in detail. The functional perovskite was characterized using different standard characterization techniques and the results were analyzed appropriately. The manuscript is well arranged, appropriately cited, and in line with the molecules journal. However, there are a few reviewer comments as follows:

1.      The reviewer suggests a revision of the manuscript in terms of grammatical and typo errors. For example:

a.       Page No. 7, Line No. 210, the sentence “Sze-Mun Lam and al [42]…” needs correction.

b.      Page No. 7, Line No. 214, “(4,34 – 239)” should be “(434 – 239)”.

c.       Table 2, correct the unit of temperature.

2.      The manuscript BaTiO3 compares well with other alternative ferroelectric cathode materials reported in the literature for MFC devices. However, the addition of a comparative table would be beneficial.

Author Response

We are very grateful for the time you took to read the manuscript and for your valuable suggestions to improve it. Herein are responses point by point to the comments:

  1. The reviewer suggests a revision of the manuscript in terms of grammatical and typo errors. For example:
  2. Page No. 7, Line No. 210, the sentence “Sze-Mun Lam and al [42]” needs correction.

It is corrected as follow:

BiFeO3-based cathodes in the form of nanoparticles synthesized by the hydrothermal method, achieved a maximum power output of 332 mW m-2, significantly lower than BaTiO3-based cathodes studied in this work [42]. 

  1. Page No. 7, Line No. 214, “(4,34 – 239)” should be “(434 – 239)”.

The values were corrected in the text (4.34 – 239).

  1. Table 2, correct the unit of temperature.

It was done.

  1. The manuscript BaTiO3compares well with other alternative ferroelectric cathode materials reported in the literature for MFC devices. However, the addition of a comparative table would be beneficial.

According to the reviewer, we have added some comparative values directly in the text, in order4 to make easier to the reader the comparison. The text was modified accordingly:

Recently, various ferroelectric photocatalytic materials have also been applied as cathodes in MFCs. BiFeO3-based cathodes, in the form of nanoparticles synthesized by the hydrothermal method, achieved a maximum power output of 332 mW m-2 [46], that is lower than the value measured for BaTiO3-based cathodes studied in this work. On the other hand, cathodes based on non-stoichiometric ferroelectric materials iso-type LiMO3 (M = Nb, Ta) modified by Cu2+, Mg2+ were investigated by some of us [19-21]; their performance as photocatalysts in air-cathode MFCs was inferior to that of the present BaTiO3 material.

Reviewer 3 Report

In this manuscript, Touach et al. reported BaTiO3 perovskite as a functional photocathode in microbial fuel cells for both power generation and wastewater treatment. This study can have implications for novel energy and environmental research. Overall, this work presents interesting results that can appeal to the readership of the journal Molecules. However, the current manuscript requires further revision to meet the publication standards. The below detailed comments need to be properly addressed.

1. The keywords need revision. Full names are suggested to be used instead of their abbreviations (MFC, COD).

2. The abbreviations for technical terms (e.g., MFC, ORR) should appear the first time the full names are given.

3. Regarding the XRD data (Fig. 1), the indexing of different planes does not match with that of the cited references (Ref. 27 and 33). JCPDS card number is not the same as given in Ref. 27. Further, please double check the space group. Is the structure a cubic (Ref. 33) or tetragonal (Ref. 27)?

4. To appeal to a broader readership, recent works about perovskite oxides and wastewater treatment are recommended to be referenced in the Introduction (e.g., ACS Sustainable Chem. Eng. 2022, 10, 1899-1909; Small Methods, 2018, 2, 1800071; Small Methods, 2022, 6, 2201099)

5. The last part of the conclusion seems like a listing of bullet points. It is suggested to re-organize them into a paragraph.

6. Please provide more experimental detail as to how the Tauc’s plot was obtained.

7. Technical terms in Tables should be defined. What are CODr in Table 1 and BOD and CE in Table 2?

Author Response

We are very grateful for the time you took to read the manuscript and for your valuable suggestions to improve it. Herein are responses point by point to the comments:

  1. The keywords need revision. Full names are suggested to be used instead of their abbreviations (MFC, COD).

It was done.

  1. The abbreviations for technical terms (e.g., MFC, ORR) should appear the first time the full names are given.

It was done.

  1. Regarding the XRD data (Fig. 1), the indexing of different planes does not match with that of the cited references (Ref. 27 and 33). JCPDS card number is not the same as given in Ref. 27. Further, please double check the space group. Is the structure a cubic (Ref. 33) or tetragonal (Ref. 27)?

-Reference 33 has been replaced by the correct reference (Journal of Nanomaterials 2015, 16, 231-231).

- The JCPDS card and indexation of the hkl plans have been corrected.

  1. To appeal to a broader readership, recent works about perovskite oxides and wastewater treatment are recommended to be referenced in the Introduction (e.g., ACS Sustainable Chem. Eng. 2022, 10, 1899-1909; Small Methods, 2018, 2, 1800071; Small Methods, 2022, 6, 2201099).

The recommended references were cited in the revised version of the manuscript.

  1. The last part of the conclusion seems like a listing of bullet points. It is suggested to re-organize them into a paragraph.

The conclusions have been rewritten.

  1. Please provide more experimental detail as to how the Tauc’s plot was obtained.

The energy gap (Eg) of was estimated through performing the UV-Vis absorption analysis. The band gap of BaTiO3 photocathode is calculated based on the Wood-Tauc method following Eq (9):

                                 (9)

Where α represents the absorption coefficient, h and ν are the plank constant and the incident light frequency respectively, A is a constant. The factor n in the equation is dependent on the type of electron transition and is either equal to 1/2 or 2 for direct and indirect band gap transitions, respectively.

  1. Technical terms in Tables should be defined. What are CODr in Table 1 and BOD and CE in Table 2?

It was done.
